# Effects of light curing on silver diamine fluoride-treated carious lesions: A systematic review

Saw Nay Min[1], Duangporn Duangthip[2], Palinee Detsomboonrat[3]*

1 Postdoctoral Researcher Program in Dental Public Health, Faculty of Dentistry, Chulalongkorn University, Bangkok, Thailand, 2 College of Dentistry, The Ohio State University, Columbus, Ohio, United States of America, 3 Department of Community Dentistry, Faculty of Dentistry, Chulalongkorn University, Bangkok, Thailand

ʘ These authors contributed equally to this work.
* palinee.d@chula.ac.th

**Data Availability Statement:** As this article is a systematic review, all pertinent data are contained within the manuscript, and there are no raw data available.

## Abstract

### Objective

This systematic review aims to evaluate the potential benefits and underlying mechanisms of combining SDF with light curing, based on available studies.

### Materials and methods

A systematic search of publications was conducted with the keywords "silver diamine fluoride" or "silver fluoride" and "dental light curing," "LED curing," "dental laser," and "dental polymerization" in 4 databases: PubMed, EBSCO, Scopus, and Google Scholar to identify English-language articles published up to March 2023. Duplicate publications were deleted. Two reviewers screened the titles and abstracts and excluded irrelevant publications. The full text of the remaining publications was retrieved. Studies investigating the effect of light-curing on SDF-treated carious lesions were included.

### Results

The 175 publications initially found included 5 laboratory studies investigating the effects of light curing on 38% SDF-treated dentine carious lesions, but no clinical study was found. Four of these studies were conducted on extracted primary teeth, and one was on extracted permanent teeth. SDF with light curing increased microhardness (n = 3, $p < .05$) showed a higher mineral density (n = 1, $p < .041$) and had more silver ion precipitation in infected dentine (n = 1, $p < .016$) compared to SDF without light curing. Moreover, no significant differences in the antibacterial activity were observed between SDF with light curing and SDF alone (n = 1, $p > .05$).

### Conclusion

Drawing from the limited number of laboratory studies, incorporating light curing subsequent to the SDF application yields potential favorable outcomes that include augmented

**Funding:** This research project is supported by the Second Century Fund (C2F), Chulalongkorn University. The funders had no role in study design, data collection and analysis, decision to publish, or preparation of the manuscript.

**Competing interests:** The authors have declared that no competing interests exist.

microhardness, elevated mineral density, and heightened silver ion precipitation within infected dentine. Future clinical research is required to confirm or refute the benefit of light curing on SDF-treated carious lesions.

## Introduction

Dental caries has remained a significant public health issue globally, with a prevalence of up to 90% in schoolchildren and adults, especially in disadvantaged populations [1]. Among the non-invasive strategies for managing caries, silver diamine fluoride (SDF) treatment has recently gained popularity due to several advantages, such as antimicrobial properties, a remineralization effect, cost-effectiveness, and accessibility for children and elders [2–4]. Studies have documented the efficacy of SDF in arresting caries on both primary [5] and permanent teeth in clinical settings [6], as well as its capacity to enhance the microhardness and mineral density of carious lesions *in- vitro* experiments [7–9].

Results demonstrated that 38% SDF is the most effective concentration in arresting caries in primary teeth [10, 11] with a biannual application demonstrating higher caries-arrest rates than an annual application [12, 13]. However, several factors can influence the efficacy of SDF, including the location of the tooth, cavity size, baseline caries experience, oral health-related behaviors, fluoride exposure, the ability to cooperate, and the application procedure [14]. Despite efforts to standardize the protocol for SDF application, the recommended protocols relied on experts' opinions, thus varying in practice [4]. Several studies have proposed that 60 seconds is the recommended duration for SDF applications; however, this may be too long for younger children, those with special needs, and older adults [4, 15].

The efficacy of SDF in managing dental caries in clinical settings has prompted dental clinicians and researchers to develop strategies to enhance its utilization [16]. In reference to a study conducted by Fung et al. [17], Crystal et al. suggested that the higher caries-arrest rates in anterior teeth compared to posterior teeth might be attributable to more natural light exposure, leading to more active silver precipitation [18]. Hence, our research question aims to investigate whether the combination of light curing and silver diamine fluoride (SDF) application enhances the effectiveness of SDF in treating carious lesions in primary teeth compared to SDF alone, by potentially accelerating the rate of silver precipitation, shortening application time, and achieving a more rapid arrest of caries. [15]. Recent ex vivo studies have highlighted that the combination of SDF and light curing application results in a considerable rise in the dentine microhardness, mineral density, and silver ion precipitation rates compared to SDF alone [15, 19, 20]. Although studies have demonstrated the potential benefits of SDF application combined with light curing on carious lesions, there remains a significant gap in the literature regarding its mechanism of action and overall effectiveness. Therefore, this systematic review aims to evaluate the potential benefits and underlying mechanisms of combining SDF with light curing, based on available studies. As SDF has been included into the Model List of Essential Medicines of the World Health Organization in 2022, alongside the growing adoption and interest in SDF [21], evidence-based protocols for SDF treatment are pivotal to ensure optimal clinical outcomes. The results of this systematic review could provide evidence to help shape future guidelines for using SDF.

## Materials and methods

This report follows the PRISMA statement (Preferred Reporting Items for Systematic Reviews and Meta-analyses) [22]. The protocol was registered in PROSPERO (CRD42023393554).

### Eligible criteria

These review questions were developed according to the population, intervention, comparison, and outcome (PICO) of the study design.

Did the addition of dental light curing after the SDF application have a significant impact on arresting caries when compared to SDF alone? In this question, the population (P) was the carious teeth, the intervention (I) was the application of dental light curing after SDF, the comparison (C) was SDF alone, and the outcomes (O) were any effects on treated teeth (such as hardness, antimicrobial properties, and penetration depth).

The criteria applied to the selection of relevant studies included: (1) studies that examined the effects of SDF with light curing in in-vivo/*in-vitro* study designs, using human dentine/dentin blocks and dentine/dentinal caries lesions in primary and permanent teeth, and (2) clinical studies evaluating the effect of SDF with light curing on carious teeth, which were compared to SDF alone.

Studies with incomplete outcome data, conference/poster presentations, case reports, opinion articles, scoping reviews, systematic reviews, and umbrella reviews were excluded.

### Information sources and search strategy

A literature search was conducted in 4 databases, SCOPUS, EBSCO, Google Scholar, and PubMed, to identify English-language manuscripts published through March 2023, using the specified keywords. The full electronic search strategy is presented in S1 Table. Two investigators (P.D. and S.N.M.) conducted the literature search.

### Selection process

EndNote (Clarivate Analytics, EndNote X8.2, 2018) was used to store and sort articles. After duplicates were eliminated, two authors separately screened the articles according to titles and abstracts to identify eligible studies. In the case of any discrepancies, full texts were examined, and the references of each relevant study were investigated to guarantee additional relevant publications. A third author (D.D.) was consulted in the event of any disagreement.

### Data collection process

The investigators independently extracted and collated data into a spreadsheet using custom-designed data extraction forms. The data included study design, number of participants or number of teeth, intervention details, treatment outcomes, instrument or methodology used for assessing outcomes, initial findings, follow-up period, study location, and final outcome; all authors agreed upon the final data to include and resolved any discrepancies through further discussion. Subsequently, the investigators checked the extracted data for accuracy.

### Assessment of the methodological quality of the included studies

The investigators independently conducted the QUIN tool (risk of bias tool for assessing *in vitro* studies conducted in dentistry) [23] to assess the quality of the included studies. The evaluation of quality was assessed based on a fixed set of bias domains (clear aims/objectives, sample size calculation, sampling technique, comparison group, methodology, operator details, randomization process, outcome measurement, outcome assessor details, blinding, statistical

analysis, and results) (Table 1). The final assessment of QUIN was performed by classifying the individual study into the level of bias risk, denoted as low, moderate, or high. To reach a consensus in any cases of discrepancy, the third reviewer was consulted.

## Results

### Study selection

A total of 175 studies were identified from the literature search in 4 databases, and 42 duplicate articles were removed. The remaining articles were screened based on the titles and abstracts. Irrelevant studies, narrative or systematic reviews, book chapters, and articles in newspapers and periodicals were also excluded. The full texts of 16 articles were retrieved and examined, and 5 articles were ultimately included in this review. The flow chart of the study is shown in Fig 1.

### Study characteristics

The included studies were all *in vitro* and conducted between 2021 and 2022 (Tables 2 and 3). Four studies used extracted primary teeth with existing dentine caries, whereas the remaining study used extracted permanent teeth with existing dentine caries. All included studies investigated the effects of light curing on SDF-treated carious teeth, compared to SDF without light curing or SDF with different light-curing protocols [15, 19, 20, 24, 25]. However, there was a

**Table 1. QUIN tool (risk of bias tool for assessing *in vitro* studies).**

| Criteria | Details |
|---|---|
| **1. Clearly stated aims/objectives** | Study should clearly state aims and/ or objectives, which should then be followed throughout. |
| **2. Detailed explanation of sample size calculation** | Details regarding method by which given sample size calculated should be clearly stated. Details regarding software program, formula, and parameters used for calculation of sample size should also be specified. |
| **3. Detailed explanation of sampling technique** | Details regarding predefined population from which sample has been selected. Details of sampling technique and inclusion and exclusion criteria should be clearly stated. |
| **4. Details of comparison group** | Details of comparison group (positive control, negative control, or standard) should be clearly specified. |
| **5. Detailed explanation of methodology** | Clarity of procedure, method of standardization, and details of any universal standards used (if applicable) should be clearly stated. |
| **6. Operator details** | Number of operators and details regarding training and calibration of operator/s (inter-operator and intra- operator reliability) should be clearly specified. |
| **7. Randomization** | Details regarding sequence generation and allocation concealment should be clearly stated. |
| **8. Method of measurement of outcome** | Clarity of procedure and rationale for choosing method should be stated. Method of standardization along with details of any universal standards used (if applicable) should also be clearly specified. |
| **9. Outcome assessor details** | Number of outcome assessors and details regarding training and calibration of assessor/s (inter- outcome and intra-outcome assessor reliability) should be clearly specified. |
| **10. Blinding** | Details regarding blinding of operator(s), outcome assessor(s), and statistician should be clearly specified. |
| **11. Statistical analysis** | Details regarding software program used and statistical analysis should be clearly specified. |
| **12. Presentation of results** | Outcome should be based on predefined aims and/or objectives. All data should be adequately tabulated with baseline data clearly specified (if applicable). |

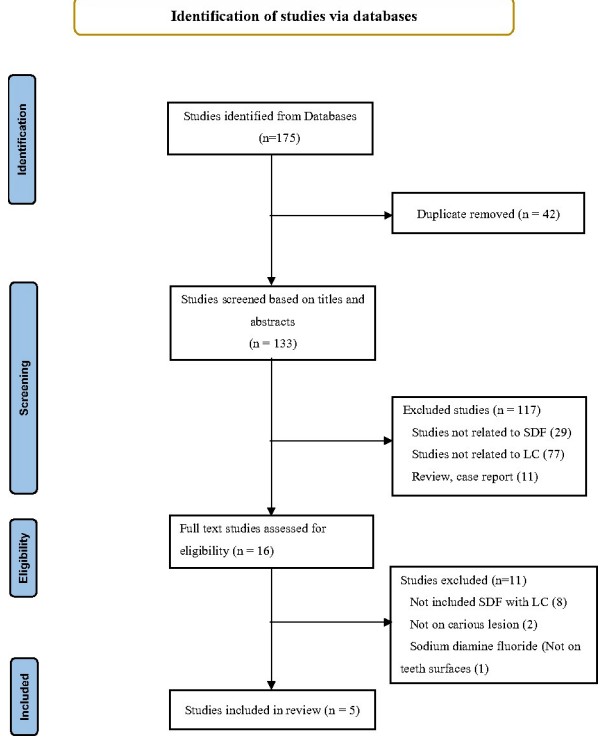

**Fig 1. Flow diagram for the systematic review.**

wide variation regarding specimen fabrication, the duration of SDF application, and the evaluation period.

## Quality assessment of the included studies

The studies included in this review exhibit a range of bias risks, encompassing low to moderate risk, as assessed by the QUIN tool. Among the 5 conducted *in vitro* studies, only 2 demonstrated a low bias risk, with QUIN scores exceeding 70% [15, 20] whereas the remaining 3 studies exhibited a moderate bias risk, as evidenced by QUIN scores falling within the 50% to 70% range [19, 24, 25]. Lack of clear reporting regarding blinding procedures was a prevalent issue across all studies [15, 19, 20, 24, 25]. The quality assessment of the studies is shown in Table 4.

## Results of the individual studies

Two studies demonstrated that the SDF with light curing resulted in a substantially higher average hardness of sound dentine beneath the infected dentine when compared to the group that used SDF alone ($p < .001$ [19 and $p < .036$) [20] respectively). This finding suggests that the addition of light curing may enhance the effectiveness of SDF in promoting dentine hardness. Nonetheless, according to Sayed et al, [25] the hardness of dentine in the group using SDF with light curing was found to be higher than the SDF-only group but only at a depth of 200 μm from the surface of the lesion. In other words, the enhancement in dentine hardness due to the light curing was observed specifically at this particular depth. Conversely, the group with SDF and the Er, Cr: YSGG laser demonstrated the highest mean dentine hardness value

**Table 2. Characteristics of the included studies.**

| Author | Study design | Agent | Type of Light cure | Inclusion criteria | No. of samples | Procedures |
|---|---|---|---|---|---|---|
| **Chanratchakool et al., 2022** | *Vitro* (Randomization) | 38% Silver Diamine Fluoride (Saforide; Japan) | LED curing light (XL 3000, 3M ESPE, St. Paul, MN, USA) | Extracted primary molar with dentine caries | Total 20 dentine blocks 5 per each group | 1. SDF application alone or followed by light curing 2. Incubation with S.mutan to form biofilms for 24 hrs |
| **Hassan et al., 2021** | *Vitro* (Randomization) | 38% Silver Diamine Fluoride (Advantage arrest, Elevate Oral Care, USA) | LEDition, Ivoclar Vivadent AG, Schaan/Liechtenstein, USA | Extracted primary molar with dentine caries | Total 30 teeth 10 per each group | Two times 1. Application of SDF after drying 30s alone or followed by LC 40s 2. After 2 weeks, same procedure |
| **Karnowakul et al., 2022** | *Vitro* (Randomization) | 38% Silver Diamine Fluoride (Saforide; Japan) | LED dental curing light (Demi plus; USA) Intensity 520 mW/cm$^2$, wavelength 450–470 nm | Extracted primary molar with dentine caries | Total 40 dentine blocks 10 per each group | 1. Specimens in artificial saliva for 1hr before SDF and LC application 2. Treated specimens in demineralization solution 4 hrs and artificial saliva 20 hrs for 7 days |
| **Sayed et al., 2021** | *Vitro* | 38% Silver Diamine Fluoride (Saforide; Japan) | Light cure with halogen (Optilux 501, 600 mW/cm$^2$, Demetron, Danbury, CT, USA) | Extracted permanent teeth with dentine caries | Total 40 specimens from 20 teeth 10 per each group | 1. 60 s SDF application left for 2 min then wash for 30 s and 2 days storage in artificial saliva 2. 60s SDF with LC 10s |
| **Toopchi et al., 2022** | *Vitro* (Randomization) | 38% Silver Diamine Fluoride (Advantage Arrest, Elevate Oral Care, USA) | Bluephase, Ivoclarvivdent, N.Y., USA Intensity 2,000 mW/cm$^2$, wavelength 385-512nm | Extracted primary maxillary incisor with dentine caries | Total 16 teeth 8 per each group | Two times 1. Application of SDF alone or followed by LC 40s 2. After 2 weeks, same procedure |

among the groups, surpassing both the SDF with light curing and SDF-alone groups.[19]. This suggests that the combination of SDF with the Er, Cr: YSGG laser may yield the most favorable dentine hardness outcomes.

Regarding antibacterial activity, Chanratchakool et al. reported no significant differences in bacterial numbers among the groups exposed to SDF, regardless of the duration of exposure or light curing [24]. Karnowakul et al. [15] found a significant difference in mean mineral density between the SDF with light curing group and that without light curing ($p < .041$); in addition, 10 s of SDF with light curing showed no difference when compared to the group with 60 s of SDF. Toopchi et al. [20] found that silver ion precipitation in infected dentine was statistically significantly higher in the case of SDF with light curing ($p < .016$). However, no statistically significant difference was noted in fluoride ion precipitation. Nevertheless, the penetration depth of SDF was greater when only SDF was applied ($p < .001$).

## Discussion

Evidence shows that SDF is effective for caries management in primary and permanent dentitions [7]. In the last few years, there has been growing interest in using SDF with the additional light curing due to its capacity to influence the penetration depth and dentine hardness of carious teeth [26]. Based on the results of the current review, no clinical trial was performed to assess the efficacy of light curing following silver-diamine fluoride treatment in arresting carious lesions in primary and permanent teeth. Despite limited information about light curing on SDF's success, this systematic review revealed the positive effects of light curing on SDF-treated carious teeth *in vitro* studies.

**Table 3. Summary of the included studies.**

| Author | Agent Comparator | Outcome | Measurement method of outcome | Results | Findings |
|---|---|---|---|---|---|
| **Chanratchakool et al., 2022** | G1:Distill water G2:10 sec SDF G3:10s SDF + 20s LC G4:60 sec LC | Antibacterial activity (Bacterial count) | Colony count CFU/ml | G1: $8.39 \times 10^6 \pm 4.94 \times 10^6$ G2: 0 G3: $284 \pm 284$ G4: 0 | There was no significant difference in bacterial count among 10s SDF, 10s SDF with LC and 60s SDF |
| **Hassan et al., 2021** | G1:30s SDF + 10s Er, Cr: YSGG G2:30s SDF + 40s LC G3:30s SDF | Dentine hardness (Microhardness) | Vickers surface hardness test (Micro met 6040, Buhler, USA) | G1: 891.24 $\pm 37.33$ kgf/mm$^2$ G2: 266.65 $\pm 90.81$ kgf/mm$^2$ G3: 117.91 $\pm 19.19$ kgf/mm$^2$ | 1. Mean difference of microhardness was highest in Laser with SDF. 2. Mean difference of microhardness of SDF with LC was significantly higher than SDF alone. |
| **Sayed et al., 2021** | G1:Control G2:Caries Check G3:60s SDF G4:60s SDF + 10s LC | Dentine hardness (Microhardness) | Vickers surface hardness test (MKV-E hardness tester, Japan) SEM-EDS (SEM: FESEM, S-4500, Hitachi, Japan, EDS: JSM-IT100 SEM, JEOL, Japan) | Hardness at 200 μm depth G1: 75.2±1.7 G2: 25.3±2.8 G3: 50.7±2 G4: 56.7±0.6 | 1. SDF groups were significantly higher in hardness than CC group. 2. Both SDF groups showed multiple crystal formation and no significant difference between SDF and SDF+LC |
| **Toopchi et al., 2022** | G1:60s SDF G2:60s SDF + 40s LC | Dentine hardness Penetration depth of SDF Ions precipitation | SEM-EDS (SEM: Amaray 3300, Bedford, Mass, EDS: Plus Tor, USA) Vickers surface hardness test (Micromet 2100, Buhler) | Penetration depth G1: 130±50 μm G2: 60±10μm Silver precipitation G1: 9.28±3.53 G2: 24.61±14.74 Hardness G1: 558.07 ±119.08 kgf/mm$^2$ G2: 702.26 ±144.6 kgf/mm$^2$ | 1. Dentine hardness was significantly higher in SDF with LC. 2. Silver ion precipitation only at infected dentine was significantly higher in SDF with LC. 3. Fluoride ion precipitation was similar in both groups |
| **Karnowakul et al., 2022** | G1:10 s 38% SDF G2:60 s 38% SDF G3:10s SDF + 20 s LC G4:60s SDF + 20s LC | Mineral density (MD) | Digital subtraction radiographic analysis (Image-Pro Plus) SEM-EDS (SEM: JSM-IT300, JEOL, Japan, EDS: Oxford instrument, High Wycombe, UK) | G1: 3.47 (5.34) G2: 6.36 (5.08) G3: 7.90 (3.51) G4: 10.47 (6.63) | 1. MD in LC group were significantly higher than those without LC. 2. Denser mineral content layers showed in LC groups than non-LC groups. |

The increased hardness of dentine indicates a more robust and more dense tooth structure of the carious lesion, which can prevent further progression [27]. Three studies [19, 20, 25] showed a marked increase in dentine hardness under the carious lesion when SDF was combined with light curing as compared to SDF alone. It can be inferred that the increase in dentine hardness is due to the high polarizing capacity of silver, which facilitates the strong bonding with oxygen, phosphorus, and sulfur groups to form a silver oxide, silver phosphate, and silver sulfide after light exposure [28]. Silver sulfide is a dense, black solid that is insoluble in all solvents and contributes to the black staining observed on surfaces treated with SDF [20]. However, Sayed et al. observed a rapid drop in Vickers hardness values (VHNs) beyond 200 μm in SDF with the light curing group in contrast to the gradual decrease of VHN values for the SDF-only group [25]. This result might be associated with the penetration of SDF due to the shorter time and rapid silver reduction induced by light curing. These studies used the VHN to measure tooth hardness due to its avoidance and easy detection of measurement errors [29]. It is essential to recognize that aspects such as sample preparation techniques, sample dehydration, amount and method of the SDF application, lesion depth, and the hardness test's location could influence the hardness test measurement *in vitro* studies [19, 30].

**Table 4. Assessment of the quality of the studies according to the QUIN tool.**

| Study | Criteria 1 | Criteria 2 | Criteria 3 | Criteria 4 | Criteria 5 | Criteria 6 | Criteria 7 | Criteria 8 | Criteria 9 | Criteria 10 | Criteria 11 | Criteria 12 | Total score | Final score % | Risk of Bias |
|---|---|---|---|---|---|---|---|---|---|---|---|---|---|---|---|
| **Chanratchakool et al., 2022** | 2 | 0 | 2 | 2 | 2 | 0 | 1 | 2 | 0 | 0 | 2 | 2 | 15 | 62.5 | Medium |
| **Hassan et al., 2021** | 2 | 2 | 2 | 1 | 2 | 0 | 1 | 2 | 0 | 0 | 2 | 2 | 16 | 66.7 | Medium |
| **Karnowakul et al., 2022** | 2 | 2 | 2 | 1 | 2 | 1 | 1 | 2 | 1 | 0 | 2 | 2 | 19 | 79.1 | Low |
| **Sayed et al., 2021** | 2 | 0 | 2 | 1 | 2 | 1 | 0 | 2 | 0 | 0 | 2 | 2 | 14 | 58.3 | Medium |
| **Toopchi et al., 2022** | 2 | 2 | 2 | 2 | 2 | 0 | 1 | 2 | 0 | 0 | 2 | 2 | 17 | 70.8 | Low |

Adequately specified = 2; Inadequately specified = 1; Not specified = 0

Final score: > 70% = Low risk of bias; 50 to 70% = Medium risk of bias; < 50% = High risk of Bias based on the formula of QUIN tool: Final score = Total score * (100) / (2 * number of criteria applicable)

Regarding the penetration of SDF on the tooth surface, Toopchi et al. [20], reported that the penetration depth of SDF into the sound dentine beyond the infected dentine was higher in the SDF-only group than in the SDF with the light curing group. Moreover, the silver ion precipitation in the SDF with light curing was significantly higher than in the SDF at the infected dentine; however, no difference was observed at affected and sound dentine. It is possible that the increased reduction of silver solution caused by light curing did not provide enough time for penetration of the carious lesion, thus resulting in less penetration depth and increased silver precipitation at the infected dentine. In contrast, an *in vitro* study found no significant difference in silver penetration depth between the groups with SDF only and SDF with light curing on a non-carious primary molar [31] The differences in the results may be attributed to the SDF penetration being affected by the extent of demineralization, sample preparation, sample dehydration, and the teeth used in the studies. Karnowakul et al. [15] observed higher mean mineral density differences (mMDD) in the SDF with light curing group compared to without. They found no significant difference in mMDD between 10 seconds of SDF with light curing and 60 seconds of SDF alone, suggesting similar effects on the remineralization process. This implies that SDF with light curing for a shorter duration could be as effective as the longer recommended application time for caries control. The potential mechanism is that dental light curing increases silver ion precipitation in the diseased dentin layer of carious lesions. This layer absorbs more light than deeper layers, with natural light acting as a catalyst in reducing the silver solution, producing silver ions and nanoparticles. Thus, dental light curing serves as an initiating agent, enabling rapid reduction and shorter SDF penetration times. Nevertheless, due to the limited number of five in-vitro studies (4 out of 5 tested on primary teeth) and a lack of clinical studies, the mechanism of using Light curing to improve the clinical outcomes is not yet fully understood. Therefore, a robust conclusion cannot be drawn.

Concerning the antibacterial properties, a recent *in vitro* study revealed no significant difference in bacterial activity between SDF-treated groups, regardless of the application time or light curing [24]. This finding was consistent with the previous study, which suggested that SDF with light curing had a similar antimicrobial action to SDF [26]. In addition, the shorter application time with light curing does not impede or diminish the antibacterial activity of the SDF as compared to the recommended approach [15].

All of the studies included in this systematic review were *in vitro* studies. As they all used the extracted natural carious teeth and attempted to simulate the oral environment, the extrapolation of the results to clinical circumstances is limited due to the complex dynamic nature of an oral environment. The included studies had a wide range of sample preparation, characteristics of the extracted teeth, exposure parameters, and evaluation periods. However, the current review of studies has demonstrated the favorable outcomes of SDF with light curing, such as increased microhardness, higher mineral density, a more significant amount of silver ion precipitation in infected dentine, no inhibition of antibacterial properties, and a potential reduction in chair time for caries management. Despite the favorable findings observed in the included *in-vitro* studies, there are some concerns regarding their susceptibility to false-positive outcomes, limited external validity, and challenges in generalizing their results to real clinical settings. Therefore, future clinical research is needed to warrant or refute the additional benefit or clinical effectiveness of light curing after an SDF application, compared to SDF without light curing. In addition, laboratory studies are necessary to comprehend the mechanism of SDF with light curing and its effect on both carious tooth surface and pulp tissues.

## Conclusion

Drawing from the limited number of laboratory studies, incorporating light curing subsequent to the SDF application yields potential favorable outcomes that include augmented microhardness, elevated mineral density, and heightened silver ion precipitation within infected dentine. Future clinical research is required to confirm or refute the benefit of light curing on SDF-treated carious lesions.

## Supporting information

**S1 Checklist.**
(DOCX)

**S1 Table. Search strategy.**
(DOCX)

## Author Contributions

**Conceptualization:** Saw Nay Min, Duangporn Duangthip, Palinee Detsomboonrat.

**Data curation:** Saw Nay Min, Duangporn Duangthip, Palinee Detsomboonrat.

**Methodology:** Saw Nay Min, Duangporn Duangthip, Palinee Detsomboonrat.

**Writing – original draft:** Saw Nay Min, Palinee Detsomboonrat.

**Writing – review & editing:** Saw Nay Min, Duangporn Duangthip, Palinee Detsomboonrat.

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
