## [Decision Letter · Decision Letter 0]

2 Apr 2024

PONE-D-23-29346Effects of light curing on silver diamine fluoride-treated carious lesions: A systematic reviewPLOS ONE

Dear Dr. Detsomboonrat,

Thank you for submitting your manuscript to PLOS ONE. After careful consideration, we feel that it has merit but does not fully meet PLOS ONE’s publication criteria as it currently stands. Therefore, we invite you to submit a revised version of the manuscript that addresses the points raised during the review process.

/>==============================

We look forward to receiving your revised manuscript.

Kind regards,

Marilia Mattar de Amoêdo Campos Velo

Academic Editor

PLOS ONE

 [This research project is supported by the Second Century Fund (C2F), Chulalongkorn University.].  

 [This research project is supported by the Second Century Fund (C2F), Chulalongkorn University.].

  [This research project is supported by the Second Century Fund (C2F), Chulalongkorn University.].

Reviewers' comments:

Reviewer's Responses to Questions

**Comments to the Author**

1. Is the manuscript technically sound, and do the data support the conclusions?

Reviewer #1: Partly

2. Has the statistical analysis been performed appropriately and rigorously? 

Reviewer #1: N/A

3. Have the authors made all data underlying the findings in their manuscript fully available?

Reviewer #1: Yes

4. Is the manuscript presented in an intelligible fashion and written in standard English?

Reviewer #1: No

5. Review Comments to the Author

Reviewer #1: The abstract lacks specific details regarding the date of data collection and the range of languages examined—whether it includes articles solely in English or those in other languages as well.

Additionally, there is an inconsistency in the presentation of authors' initials within the methodology section.

The term "in vitro" should be italicized throughout the document for proper emphasis.

Drawing a conclusion about the potentiation of antimicrobial effects by light based solely on a single referenced article is insufficient and lacks robust support.

Furthermore, in the table, distinct sections should be allocated for studies assessing microhardness, antimicrobial effects, and other effects of Silver Diamine Fluoride (SDF).

The entire text requires thorough editing to address these language and grammatical inaccuracies.

6. PLOS authors have the option to publish the peer review history of their article (what does this mean?). If published, this will include your full peer review and any attached files.

Reviewer #1: No

---

## [Author Response · Author response to Decision Letter 0]

24 Apr 2024

Response item 1: We edited the manuscript meets PLOS ONE's style requirements

Response item 3 : We edited ‘Funding Information’ and ‘Financial Disclosure’ sections 

Response item 4: We added state: "The funders had no role in study design, data collection and analysis, decision to publish, or preparation of the manuscript." in cover letter.

Response item 5: We removed any funding-related text from the manuscript

Reviewers' comments:

Reviewer #1: The abstract lacks specific details regarding the date of data collection and the range of languages examined—whether it includes articles solely in English or those in other languages as well.

Response: We added the date of data collection and the range of languages examined in the abstract (Page 1 Line 8-9).

Additionally, there is an inconsistency in the presentation of authors' initials within the methodology section.

Response: We checked the presentation of authors' initials within the methodology section consistently. 

The term "in vitro" should be italicized throughout the document for proper emphasis.

Response: We changed "in vitro" to be “in vitro” throughout the document.

Drawing a conclusion about the potentiation of antimicrobial effects by light based solely on a single referenced article is insufficient and lacks robust support.

Response: We understand and acknowledge your point that relying solely on one reference may not provide robust support for our conclusion. Upon reflection, we agree that a more comprehensive analysis incorporating multiple studies would enhance the credibility and robustness of our findings. To address this concern, we have revised the conclusion section to regard the potential potentiation of antimicrobial effects by light in conjunction with SDF treatment, based solely on a single referenced article. It also underscores the importance of further research to validate these findings in clinical settings (Page 16 Line 223-227).

Furthermore, in the table, distinct sections should be allocated for studies assessing microhardness, antimicrobial effects, and other effects of Silver Diamine Fluoride (SDF).

Response: In Table 3, distinct sections have been allocated for studies assessing the micro hardness, antimicrobial effects, and other effects of Silver Diamine Fluoride (SDF) in outcome column. However, the author has reordered it so that studies with similar outcomes are grouped.

The entire text requires thorough editing to address these language and grammatical inaccuracies.

Response: The entire text was edited to address these language and grammatical inaccuracies by the Papercheck. We also attached the certificate of editing.

And We uploaded our figure file to PACE digital diagnostic tool and edited.

---

## [Editor Report · Decision Letter 1]

1 May 2024

PONE-D-23-29346R1Effects of light curing on silver diamine fluoride-treated carious lesions: A systematic reviewPLOS ONE

Dear Dr. Detsomboonrat,

Thank you for submitting your manuscript to PLOS ONE. After careful consideration, we feel that it has merit but does not fully meet PLOS ONE’s publication criteria as it currently stands. Therefore, we invite you to submit a revised version of the manuscript that addresses the points raised during the review process.

We look forward to receiving your revised manuscript.

Kind regards,

Marilia Mattar de Amoêdo Campos Velo

Academic Editor

PLOS ONE

Additional Editor Comments:

The authors have adressed all comments by the reviewer. However, it is still unclear the main research question of this SR.

Reviewers' comments:Silver diamine fluoride (SDF) has been proven to be a very useful and effective strategy to treat carious lesions, in particular involving primary teeth. Clinical trials have been conducted all around the world and evidence is robust to indicate this product as an anticariogenic strategy.

The purpose of this study was focused in analyze the effect of its combination with light curing as a tool that could enhance its effective, through a systematic review (SR). Only in vitro studies were found according to the criteria, in which the majority (4 of 5) was tested on primary teeth. Based on the available evidences up to now, this study lacks to show a consistent argument to this combination. The observation that the light could improve, with no additional previous evidence of the mechanism of action and the lack of a consistent conclusion, turns this study fragile. However, authors justify this use because of some studies as references 15, 19 and 20. In this first study, Karnowakul et al. 2022 state that the improved performance of the SDF could be attribute to “When the silver particles are exposed to light, the shape of the particles tends to change from spherical to plate-like triangular, square, or hexagonal”. However, the mechanism is still not explained. In the second study, the mechanism of the combination is attributed to laser in a different condition, since the laser is used in particular to dentin condition and not towards the SDF. In the third one, the authors use methods that show more mineralized condition but once again only speculate that the light can be useful.

Authors claims that there is no (SR) related to this subject, but it is not quite an argument that it need to be done.

Please, the research question needs to be clearly.

---

## [Author Response · Author response to Decision Letter 1]

6 Jun 2024

Dear Reviewer,

We have addressed your concerns and made revisions accordingly. Below is our detailed response to each of the points raised:

1. Purpose and Research Question

Reviewer Comment: The research question needs to be clearly stated.

Response: Thank you for pointing this out. We have revised the manuscript to clearly articulate our research question. The research question now reads: "Does the combination of silver diamine fluoride (SDF) and light curing enhance the effectiveness of SDF in treating carious lesions in primary teeth compared to SDF alone?" This has been updated in the Introduction section in Line 74-77

2. Lack of Consistent Argument and Mechanism of Action

Reviewer Comment: The study lacks a consistent argument for the combination of SDF and light curing. The mechanism by which light curing could improve the effectiveness of SDF is not well-explained, and the conclusions drawn are not robust.

Response: We expand the Discussion section to include a more comprehensive review of the potential mechanisms and add references. Specifically, we discuss the hypothesis that light exposure could alter the shape and activity of silver particles, as suggested by Karnowakul et al. (2022). We also elaborate on the possible interactions between light and SDF that might contribute to increased mineralization and antimicrobial activity. Nevertheless, due to the limited number of studies with a lack of clinical studies to support the effectiveness, we acknowledge that the mechanism behind is not yet fully understood and a robust conclusion cannot be drawn.

3. Justification for Systematic Review

Reviewer Comment: Authors claim that there is no systematic review (SR) related to this subject, but this alone is not a sufficient justification for the study.

Response: We have revised the Introduction to better justify the significance of our systematic review in Line 81-87. We emphasize the clinical relevance of exploring adjunctive treatments to enhance the efficacy of SDF, the growing interest in non-invasive dental treatments, and the potential implications for dental care. We also highlight the gap in the literature regarding the combination of SDF and light curing, which our study aims to address.

4. In Vitro Studies and Evidence Robustness

Reviewer Comment: Only in vitro studies were found according to the criteria, with the majority (4 out of 5) tested on primary teeth. The study lacks robust evidence for the combination of SDF and light curing.

Response: Agree, we add your valuable comment to the manuscript in line 235-237. “due to the limited number of five in-vitro studies (4 out of 5 tested on primary teeth) and a lack of clinical studies, the mechanism of using Light curing to improve the clinical outcomes is not yet fully understood. Therefore, a robust conclusion cannot be drawn.”

We have clarified the limitations of our systematic review, particularly the reliance on in vitro studies, in the Limitations section in Line 251-252. We discuss the implications of these limitations for the generalizability of our findings to clinical practice and emphasize the need for future clinical trials to establish robust evidence in Line 252-256. Additionally, we have provided a more detailed summary of the in vitro studies included, highlighting their methodologies, findings, and relevance to the research question in line258-261.

5. References and Specific Studies

Reviewer Comment: Specific studies (references 15, 19, and 20) are cited, but their relevance and mechanisms are not well explained.

Response: We have revisited the referenced studies and provided a more detailed discussion of their findings and relevance to our research. In the revised manuscript, we explicitly discuss how each study contributes to our understanding of the potential benefits and mechanisms of combining SDF with light curing in Line 203-205, 217-230. We also critically analyze the limitations of these studies and how they inform our systematic review in Line 230-237.

We hope that these revisions satisfactorily address your concerns. We believe that the changes made have strengthened our manuscript, and we appreciate the opportunity to improve our work based on your insightful feedback.

Sincerely,

---

## [Editor Report · Decision Letter 2]

16 Jun 2024

Effects of light curing on silver diamine fluoride-treated carious lesions: A systematic review

PONE-D-23-29346R2

Dear Dr. Detsomboonrat,

We’re pleased to inform you that your manuscript has been judged scientifically suitable for publication and will be formally accepted for publication once it meets all outstanding technical requirements.

Kind regards,

PLOS ONE

---

## [Editor Report · Acceptance letter]

27 Jun 2024

PONE-D-23-29346R2 

PLOS ONE

Dear Dr. Detsomboonrat, 

I'm pleased to inform you that your manuscript has been deemed suitable for publication in PLOS ONE. Congratulations! Your manuscript is now being handed over to our production team.

Kind regards, 

on behalf of

Dr. Marilia Mattar de Amoêdo Campos Velo 

Academic Editor

PLOS ONE